# TE-TransReID: Towards Efficient Person Re-Identification via Local Feature Embedding and Lightweight Transformer

**DOI:** 10.3390/s25175461

**Published:** 2025-09-03

**Authors:** Xiaoyu Zhang, Rui Cai, Ning Jiang, Minwen Xing, Ke Xu, Huicheng Yang, Wenbo Zhu, Yaocong Hu

**Affiliations:** 1School of Electrical Engineering, Anhui Polytechnic University, Beijing Road No. 8, Wuhu 241000, China; 2240342204@stu.ahpu.edu (X.Z.); 19155387996@163.com (R.C.); 3240201112@stu.ahpu.edu.cn (M.X.); kexu@ahpu.edu.cn (K.X.); hcyang@ahpu.edu.cn (H.Y.); wbzhu@ahpu.edu.cn (W.Z.); 2College of Computer and Control Engineering, Northeast Forestry University, Harbin 150040, China; jiangning@nefu.edu.cn

**Keywords:** person re-identification, lightweight transformer, efficient feature-fusion modules, trade-off

## Abstract

**Highlights:**

**What are the main findings?**
We use a two-branch lightweight feature-extraction network.A lightweight and efficient fusion module is constructed.

**What are the implications of the main findings?**
The overall network is lightweight and features are comprehensively processed.Characteristics of various forms are integrated without significantly increasing the number of parameters.

**Abstract:**

Person re-identification aims to match images of the same individual across non-overlapping cameras by analyzing personal characteristics. Recently, Transformer-based models have demonstrated excellent capabilities and achieved breakthrough progress in this task. However, their high computational costs and inadequate capacity to capture fine-grained local features impose significant constraints on re-identification performance. To address these challenges, this paper proposes a novel Toward Efficient Transformer-based Person Re-identification (TE-TransReID) framework. Specifically, the proposed framework retains only the former L-th layer layers of a pretrained Vision Transformer (ViT) for global feature extraction while combining local features extracted from a pretrained CNN, thus achieving the trade-off between high accuracy and lightweight networks. Additionally, we propose a dual efficient feature-fusion strategy to integrate global and local features for accurate person re-identification. The Efficient Token-based Feature-Fusion Module (ETFFM) employs the gate-based network to learn fused token-wise features, while the Efficient Patch-based Feature-Fusion Module (EPFFM) utilizes a lightweight Transformer to aggregate patch-level features. Finally, TE-TransReID achieves a rank-1 of 94.8%, 88.3%, and 85.7% on Market1501, DukeMTMC, and MSMT17 with a parameter of 27.5 M, respectively. Compared to existing CNN–Transformer hybrid models, TE-TransReID maintains comparable recognition accuracy while drastically reducing model parameters, establishing an optimal equilibrium between recognition accuracy and computational efficiency.

## 1. Introduction

Person ReID encompasses the technology of accurately identifying and matching individuals of the same identity across diverse surveillance scenarios or multiple camera viewpoints. It constitutes a critical research area within computer vision, particularly in the field of intelligent security. Its core functionality lies in extracting features of individuals from images captured under varying shooting conditions and viewpoints, thereby enabling the identity comparison of individuals across scenarios or under varying poses within the same scene [1,2]. Ideally, person ReID technology should be capable of addressing the occlusion of individuals, variations in illumination, postural variations, and other challenges present in large volumes of image and video data [3,4,5]. However, “Lightweight” specifically refers to a significant reduction in model parameter count and computational complexity, enabling adaptation to deployment on resource-constrained edge devices. As illustrated in Figure 1, existing hybrid models (e.g., FusionReID) generally have a parameter count exceeding 150 M, while pure Transformer models (e.g., TransReID) also reach over 86 M. In contrast, the lightweight defined in this paper requires the parameter count to be controlled within 40 M (27.5 M for TE-TransReID), ensuring that inference speed meets the requirements of real-time surveillance scenarios. “High accuracy” refers to the model’s recognition performance on mainstream datasets reaching or approaching the current state-of-the-art level, with core metrics including rank-1 accuracy and mAP. Specifically, on benchmark datasets such as Market1501, high accuracy requires achieving a rank-1 accuracy of no less than 90% (94.8% for TE-TransReID), while maintaining stable performance in complex scenarios such as occlusion and pose variations, with no significant discrepancy compared to models with over 100 M parameters (e.g., 96.0% for FusionReID). Therefore, this paper focuses on how to achieve a true lightweight model while ensuring high accuracy, exploring the optimal balance strategy between the two.

Traditional person ReID is mainly implemented through two stages: manual feature extraction and metric learning. The hand-crafted feature extraction mainly focuses on the specific attributes of the person image, such as color distribution, clothing texture, and body structure proportion. For example, color features are described by extracting the HSV color-space histogram of person images [6]. In addition, the metric learning algorithm is used to learn a suitable distance measurement space; that is, after processing, the cosine distance of the same person image in this space is very close, and the distance of different images is very far, so as to realize the ReID of personnel. Representative algorithms of this method include the SDALF [7].

With the wide application of deep learning technology in the field of computer vision, the implementation of deep learning algorithms in person ReID research has shown great potential. In recent years, researchers have constructed a large number of CNN [8,9,10] and Transformer [11,12,13] based network models, and used large-scale public person-image datasets: Market1501 [14], DukeMTMC [15], MSMT17 [16] for end-to-end training and learning, so that the model can automatically learn to extract local or global person features from original images. The network under deep learning approaches is mainly divided into CNN-based approaches, Transformer-based approaches, and the hybrid CNN–Transformer-based approaches; the CNN network mainly extracts local features (PCB, MGN and so on [17,18]), and the Transformer method mainly extracts the global semantic features of the image for matching, as its global modeling and long-distance dependent capture capabilities are more suitable for problems such as person occlusion [19,20]. The fusion method combines the advantages of CNN and Transformer [21,22,23]; most of them can take into account both global and local characteristics, and can better avoid information loss.

Despite the rapid advancements in Person ReID and the development of numerous algorithmic networks in recent years, inherent limitations persist across different network paradigms. Specifically, CNN-based methods exhibit suboptimal performance under occlusion, while Transformer-based approaches suffer from low computational efficiency and high complexity. The cumulative effect of their multi-layer self-attention mechanisms incurs excessive computational overhead, and their performance degrades when handling partial occlusion or significant pose variations. For example, FusionReID achieves high recognition accuracy via parallel CNN-Transformer frameworks with extensive comparative operations; however, the stacking of Vision Transformer blocks within its architecture leads to heightened network complexity [24]. That is, it achieves a rank-1 accuracy of 95.9% on Market1501 with a parameter quantity of 153.8 M.

In view of the above problems of different methods, this paper proposes a new feature-fusion network named TE-TransReID, which uses lightweight fusion and extraction modules to achieve a better balance between matching computational efficiency and accuracy. The contributions of this paper are as follows:(1)This paper proposes a lightweight CNN–Transformer network architecture TE-TransReID. As shown in Figure 2, it uses lightweight CNN to replace global feature extraction. Both architectures use only the first *L’* layers (*L* stands for the total number of layers in the original architecture, and N stands for the number of layers not needed during processing) to gradually extract features, which greatly reduces the number of network parameters and calculation.

(2)For feature fusion, we design two lightweight and efficient fusion blocks: a feature-fusion module—ETFFM—for feature token vectors and a lightweight feature-fusion module—EPFFM—for feature patches.(3)Compared with the classical method TransReID, and the newer method FusionReID, TE-TransReID optimizes the computational performance of the network with good accuracy, and realizes more lightweight parameters. Figure 1 shows that we are able to achieve a level of accuracy comparable to other large-parameter models with very small model parameters compared to other methods [8,11,17,24,25,26,27].

## 2. Related Work

In this paper, we first briefly introduce the mainstream methods and research directions of person ReID in recent years, and explain our insights and some related explorations on current development and research. In the following article, we will discuss CNN, Transformer, and their hybrid networks.

### 2.1. CNN-Based Person ReID

CNN convolutional neural networks are good at capturing local fine features such as a person’s clothing and torso pose) and spatial structure information, which makes this method popular in computer vision: Spindle Net, PPCL, DSA-ReID, and other networks realize the learning of correspondence between image patches [28,29,30,31]. However, CNNs exhibit a weak ability to model global contextual relationships (struggling to capture associations between distant pixels) and also tend to overemphasize local information, which ultimately results in feature redundancy. It can be seen that the CNN method alone still faces many challenges in developing the dynamic scene and information.

### 2.2. Transformer-Based Person ReID

Common Transformer architectures use end-to-end training and a self-attention mechanism to capture long-distance dependencies, avoiding problems such as excessive computation and limited local receptive field that may occur in CNN networks. For example, the PSTR [32] framework proposed by Jiale Cao et al. and the TransReID method used by Shuting He [11] have demonstrated good data performance on the public dataset. However, there are some drawbacks to this approach: global detail information may be ignored, and some features may not be aligned when fused, resulting in mismatching.

In summary, global feature extraction takes the entire image as the processing unit, capturing global semantic correlations and long-range dependencies through encoding the overall information of the image. Local feature extraction, by contrast, focuses on local regions within the image (such as the head, torso, and limbs), capturing fine-grained information through refined processing of local details. The two differ in their extraction scope and the granularity of information they process, which leads to their distinct applicable scenarios. Consequently, in the field of person re-identification, numerous studies currently seek to fuse these two types of features to achieve superior performance.

### 2.3. Hybrid CNN-Transformer-Based Person ReID

Feature fusion is generally to balance the processing relationship between coarse-grained and fine-grained features, so that the model can have stronger robustness and the ability to deal with complex scenes. In general, the main methods are to use global features to provide overall context information, and combine local features to supplement the details. Architectures like HAT, CMT, and Conformer all focus on coarse-grained fusion [20,33,34]. In addition, Yuhao Wang et al. recently proposed the FusionReID method [24], which processes images separately using parallel CNN and Transformer branches, and achieves feature fusion by stacking CTM (dual-attention hybrid architecture) for person matching [35,36]. In other words, numerous existing methods adopt multi-branch network architectures or complex feature-fusion modules, which ultimately elevates the difficulty of model training and optimization.

So, this paper designs a parallel-processing person ReID network based on MobileNetV2 and Vision Transformer, after a lot of learning of the former [37,38]. However, compared with the traditional two-branch CNN–Transformer fusion framework, the branch architectures we use are networks with small parameters (that is, this paper makes a large contribution to constructing a lightweight model), and two more efficient modules are used in the fusion stage. Among them, the patch-based feature fusion uses lightweight operations under Transformer, and these processes make the architecture parameters of this paper smaller. All of them still maintain excellent accuracy in person ReID.

The experimental results show that the proposed lightweight ReID network TE-TransReID can use a more lightweight architecture and fewer Vision Transformer layers than FusionReID network at a relatively flat level of precision and accuracy [11,24]. Additionally, adaptive innovation is carried out on the gate-level structure, and the mutual–gated structure and lightweight attention mechanism are used to process the token-level and patch-level features, respectively. In addition, taking Market1501 dataset as an example, its performance is not inferior to that of current ReID methods, and the number of parameters is smaller than that of FusionReID, showing the significant advantages of TE-TransReID.

## 3. Methodology

Here, an overview of the proposed person ReID framework is shown in Figure 3. This framework leverages the complementary strengths of CNN and Transformer, where the CNN network processes local feature information, while the Vision Transformer network handles global feature information [39,40]. Specifically, TE-TransReID introduces a lightweight CNN and Transformer complementary mechanism within the framework to extract person-image features; ETFFM and EPFFM aim to integrate various forms of token and patch-level features. By using a camera and position-embedding information, the early layers of the feature-extraction network process images to achieve a lightweight design.

Given an input image $I\in R^{W\times H\times C}$ (*W* (Width) and *H* (Height) indicate the spatial dimensions of the feature map. During the processing of input images by MobileNetV2, the original spatial size of the input image is gradually compressed through convolution, pooling, and other operations. The resulting *W* and *H* of the feature map reflect the spatial distribution information of local regions in the image. *C* (Channel) represents the dimensionality of the feature map in the channel dimension, reflecting the richness of features extracted by the model. In MobileNetV2, the “Inverted Residual Block” is used to expand and compress feature dimensions. The value of C corresponds to the number of channels of the feature after expansion or compression), the CNN branch extracts local features through the first M layers of MobileNetV2 network and obtains patch feature $F_{local}^{M}\in R^{(W/7)\times (H/7)\times C_{l}}$. Then, we have the pooled $f_{local}^{M}\in R^{1\times 1\times C_{1}}$. The global features obtained by the Transformer branch network are extracted to obtain $F_{global}^{L}\in R^{(W/16)\times (H/16)\times C_{g}}$ and $f_{global}^{L}\in R^{1\times 1\times C_{g}}$ in the same way. ETFFM is a fusion module for token-level features. It uses a mutual–gated structure to dynamically fuse $f_{global}^{L}$ and $f_{local}^{M}$ into $f_{Token}$. EPFFM is aimed at patch-level features, using a lightweight Transformer and a self-attention mechanism to fuse $F_{local}^{M}$ and $F_{global}^{L}$ into $F_{Patch}$. Finally, the pooling is $f_{Patch}$; together with $f_{Token}$, it is sent to the loss function, ID Loss, for the iterative optimization of the network.

The functionalities of each component in Figure 3 are elaborated as follows:
(a)The pretrained MobileNetV2 branch serves as a lightweight convolutional feature extractor, outputting the local feature map $F_{local}^{M}$ and the local feature vector $f_{local}^{M}$ to capture fine-grained visual details.(b)It converts the input image into a sequential representation, leverages multi-layer Transformers to model long-range dependencies, and outputs the global feature map $F_{global}^{L}$ and the global feature vector $f_{global}^{L}$, thereby strengthening the modeling of semantic correlations.(c)The module in (c) targets vector-level feature fusion: $f_{local}^{M}$ and $f_{global}^{L}$ are first normalized, then fed into fully connected layers coupled with a gating mechanism that dynamically learns fusion weights between global and local features. The resultant fused feature is optimized via ID Loss and TriLoss.(d)The module in (d) enables feature map-level fusion: it concatenates $F_{local}^{M}$ and $F_{global}^{L}$, incorporates camera embeddings to supplement domain-specific information, aggregates patch-level correlations using separable self-attention, and refines features through a convolution–residual structure, with dual losses (ID Loss and TriLoss) guiding optimization.

This architecture establishes feature complementarity via the “convolution (local) + Transformer (global)” dual-branch design, achieves multi-granularity feature aggregation through the “gated vector fusion + lightweight Transformer-based feature map fusion” dual-path mechanism, and enhances domain adaptability via camera embeddings. Ultimately, it generates more comprehensive and discriminative feature representations for person re-identification, while ensuring efficiency and generalizability through lightweight components.

### 3.1. CNN for Local Feature Extraction

In this paper, we use MobileNet—MobileNetV2 as one of the backbone networks for image processing [37]. It mitigates issues such as large model size and low training/inference speed, which are incurred by stacking deep residual blocks in ResNet—a commonly used CNN backbone for local feature extraction in person ReID.

As illustrated in Figure 4, the inverted residual structure of MobileNetV2, breaks away from the design paradigm of traditional residuals and adopt a new way of working: input features (with *M* channels) first undergo convolution-based dimension expansion (with a dimension expansion ratio t, resulting in *tM* channels). This dimension augmentation prevents premature compression of features, thereby better preserving the richness of fine-grained information. Subsequently, a depthwise separable convolution is incorporated, which only models spatial correlations within individual channels, significantly reducing computational overhead. Finally, a 1 × 1 convolution is used to reduce the dimension back to *M* (ensuring consistent input and output channel counts to satisfy residual connections). By enhancing feature expression through dimension expansion and optimizing efficiency via depthwise separable convolution, this design enables MobileNetV2 to extract high-quality local features at a lightweight cost, laying a foundation for efficient feature extraction in the convolutional branch of TE-TransReID and supporting the performance of subsequent multi-module fusion. The operation of the dimensional expansion phase can be regarded as:(1)$$U=ReLU6(W_{\mathrm{expand}}\cdot I)$$(2)$$DWSConv(U)=\sum_{C=l}^{C_{expand}}K_{c}^{\left(3\times 3\right)}*U_{c}$$(3)$$F_{Local}^{M}=ReLU6(W_{proj}DWSConv(U))$$
where $W_{\mathrm{expand}}$ represents the convolution weight in the expansion operation part; ReU6 represents the processing of feature information to obtain the dimensioned information *U*; DWSConv represents depthwise separable convolution; $K_{c}^{\left(3\times 3\right)}$ represents the 3 × 3 depth convolution kernel of channel *c*; $W_{proj}$ represents the convolution weight of the projection operation; and the “*” represents the convolution operation in the model.

Equation (3) corresponds to the projection operation stage in the inverted residual structure of MobileNetV2, with its derivation based on the combined design of dimension expansion and DWSConv. After the dimension expansion in Equation (1) (increasing the number of input channels *M* to *t* × *M* via 1 × 1 convolution, where t denotes the dimension expansion ratio and $W_{\mathrm{expand}}$ represents the convolution weights for the expansion operation) and the DWSConv in Equation (2); Equation (3) maps the high-dimensional features U back to the dimension consistent with the number of input channels through 1×1 convolution. This derivation process, which first elevates dimensions to capture rich features and then reduces dimensions to compress parameters, significantly lowers computational complexity while preserving feature expressiveness. In contrast to the “dimension reduction followed by expansion” in traditional residual structures, it is more suitable for lightweight network design.

Therefore, the overall operation of the MobileNetV2 can be described as above. On the other hand, V2 introduces a linear bottleneck layer, so that it reduces the use of nonlinear activation functions in low-dimensional space, which can better avoid information loss and further reduce the computational complexity. As a result, the overall operation of the MobileNetV2 can be the same as described above. In addition, in order to facilitate the subsequent feature-fusion processing, $f_{local}^{M}$ is obtained by vectorizing the features after model extraction.

Sum up, in the image-processing stage based on CNN branches, select the V2 network with fewer model parameters, which can not only achieve a certain accuracy requirement in the amount of lightweight parameters, but also obtain $F_{local}^{M}$ and $f_{local}^{M}$ for subsequent feature processing.

### 3.2. Transformer for Global Feature Extraction

In parallel with CNN as a branch of global feature extraction, this paper designs a Transformer-based network framework. Learning from the supplementary information of methods such as [21,33,39], this paper also adopts a similar information-processing method. Before each patch feature is fed into the Transformer layer, the input-embedding sequence *P* = [*P_1_*, …, *P_N_*], the person’s position information and the camera viewpoint information are jointly supplemented with corresponding information and encoded by the L-1 layer Transformer. After a series of processes, the following can be obtained:(4)$$P_{L-1}=LayerNorm(P_{L-2}+MAS(P_{L-1}))$$(5)$$F_{global}^{L}=R_{e}\cdot Transformer^{L-1}[P_{e}+C_{e}+Flatten\cdot Patch(I)]$$
where MSA is the extraction of information, and the processing of multi-head self-attention mechanism is tested on a different $Transformer^{L-1}$; $R_{e}$ represents the vector reshaping to obtain the patch feature; $P_{e}$ and $C_{e}$ represent the position and camera embedded information; *I* represents the input image; and Flatten⋅Patch represents the segmentation and flattening of the input into a one-dimensional vector. After this paper selects the four layers, the ViT-Small model is used, which has low parameters and fewer layers but can maintain good accuracy.

First, the *I* is divided into several fixed-size image patches via the patch-embedding operation and flattened into a 1D vector sequence, which is then fused with positional embedding information and camera viewpoint-embedding information to supplement contextual details regarding spatial positions and cross-camera domains. The fused sequence is encoded by the *L-1* layer Transformer, capturing global semantic correlations and long-range dependencies through the self-attention mechanism. Finally, the encoded features undergo vector reshaping via the Reshape operation, yielding the final patch-level feature $F_{global}^{L}$.

In addition, for lightweight architecture design, TE-TransReID does not process all image data in all layers of the selected model using the usual methods [10,16,40], but implements the feature extraction, by processing the front part of the layer according to the image-processing situation in the algorithm. One of them is the *CLS Token*, which is used to generate $f_{global}^{M}$. Other token features are used for generate $F_{global}^{M}$, both of which are used in different feature fusions. The redundant processing of the latter Transformer layer is omitted, and the parameter redundancy caused by multi-head attention and layer accumulation common in the Transformer model is avoided, so as to reduce the calculation of its processing and screening features while meeting the experimental accuracy requirements.

### 3.3. EPFFM

Thirdly, according to the feature-processing method of the patch block, the use of the EPFFM is proposed in order to be able to fully consider different information in the low and high dimensions. For the fusion module, we also consider using lightweight methods as much as possible; in this paper, the patch-level features use feature-fusion network architecture based on the Transformer [18,20,23].

Let the input-feature matrix be *X* (where k denotes the length of the feature sequence and d represents the feature dimension). First, linear transformations are applied to generate the query matrix ${Q=\mathrm{XW}}_{x}$, key matrix ${K=\mathrm{XW}}_{k}$, and value matrix ${V=\mathrm{XW}}_{v}$ (with $W_{x}$, $W_{k}$, $W_{v}$ as learnable linear transformation weights, and *d* as the transformed dimension). Then, context scores *S* are computed, modeling the correlation strength between different positions via scaled dot-product attention. Subsequently, a context vector is generated by performing a weighted sum of the value matrix based on the context scores, focusing on key semantic information. Finally, after enhancing nonlinear expression through the ReLU activation function, an element-wise multiplication is performed with the value matrix *V* to obtain the output feature *Y*.

Compared with token vector processing, we directly stitch and integrate information to map blocks to obtain more comprehensive and richer information. After the convolution operation, the features are directly entered into the separable self-attention mechanism module (SSA). It is an improved self-attention mechanism, which can not only reduce the computational complexity, but also improve the efficiency. Its structure is shown in Figure 5. It mainly calculates the separated attention through different dimensional channels, and mainly reduces the amount of calculation by this process, so as to realize efficient reasoning and training. In that regard, the context scores in SSA can be viewed as follows:(6)$$\begin{array}{c} S=softmax(\frac{QK^{T}}{\sqrt{d}}),\; \end{array}S\in R^{N\times K}$$
where *S* represents the correlation between the different contextual semantics, *Q* is the query matrix obtained by another learnable linear transformation, $\sqrt{d}$ indicates a zoom operation and *K* represents the generation of context vectors [38].

The SAA compresses the attention computing space through context vector and realizes efficient feature selection and fusion by using the context score, which is both efficient and discriminative. Its core idea is to use a small amount of global semantics to guide local feature aggregation, avoid redundant calculation, and finally obtain:(7)$$Y=X+SV=X+S\times ReLU(XW_{V})$$
where $W_{V}$ indicates that the value is a learnable matrix. However, the processed features are also in a high-dimensional state. In order not to increase the calculation complexity of the model, this paper still pools $F_{patch}$ into $f_{patch}$ after fusion in this module.

The pooling operation can aggregate local features and reduce the number of features. This operation not only realizes the conversion between high and low dimensions (avoiding the introduction of noise and redundant information as much as possible), but also reduces the model’s excessive dependence on local details and pays more attention to the overall feature information of the picture. In addition, the loss function also needs to receive fixed dimension input features. The dimension reduction process of patch feature blocks is also to unify all the above feature dimensions, which is easy to be fed into the loss function for loss calculation and backpropagation.

### 3.4. ETFFM

Since the Transformer network model splits the original people image into multiple patches of the same size and then processes each patch in batches, we obtain the class token vector after extracting features through the global branch, and then capture the global dependency based on the local vector and through the multi-head self-attention mechanism. So, the first thing that we obtain are the characteristics of the token form of local and global convergence. However, the CNN branch mostly appears in the form of a feature map ($F_{global}^{L}$) after processing; that is, we need to merge it with the features extracted by the Transformer after the pooling operation into the form of a token amount.

In this paper, we propose the ETFFM for this fusion process. Based on the former, the gated-based mechanism dynamically adjusts and fuses the feature data structure; that is, we use multi-layer neural networks to form a mutual–gated structure for token features extracted from different branches [41,42]. The main principle of this structure is to normalize the two token vectors ${f^{\prime }}_{i}$ ($f_{global}^{M}$ $f_{global}^{L}$) to obtain ${{\dot{f}}^{\prime }}_{l}$, in which *i* represents global and local token features:(8)$${{\dot{f}}^{\prime }}_{l}=LayerNorm\left({f^{\prime }}_{l}\right)$$

Then, information enhancement and integration are carried out to obtain f_c_. Then, the two tokens are processed separately and cross-interchanged between gated gates. Under the fully connected layer and activation function, gated weight $G_{i}$ can be obtained:(9)$$G_{i}=\sigma (\varpi_{i}[{\dot{f}}_{local}^{M}\left\|{\dot{f}}_{local}^{L}\right.])$$
where we use “${\dot{f}}_{local}^{M}\left\|{\dot{f}}_{local}^{L}\right.$” to realize the concatenation operation, and then combine this with the learnable parameters to $\omega_{i}$ obtain $G_{i}$ jointly. The module can also dynamically adjust the weight of learning features to supplement and fuse itself with another branch feature:(10)$$f_{i,j}=G_{l}⊙{\dot{f}}_{local}^{M}\left\|{\dot{f}}_{local}^{L}\right.⊙G_{g}$$
where $f_{i,j}$ indicates that the two branches complement each other and integrate their features. Finally, the initial fusion processing of token and the processed $f_{l}$, $f_{g}$ and $f_{c}$ is carried out again to obtain $f_{Token}$:(11)$$f_{Token}=Mutual\cdot Gated(f_{i,j},f_{j,i},f_{c})$$

At this point in time, the information interaction of features at the vector level is very sufficient, and this method is more efficient and satisfactory than the traditional direct combination of global and local features.

### 3.5. Loss Function

The $f_{Token}$ and $f_{patch}$ features, after the two fusion processes, are processed using the same loss function. Since the above operations have processed different forms of fused features into token vectors of the same dimension, they can be directly input into the loss function for use in the downstream processing. In this paper, the traditional Identification Loss and TriLoss are used [3,30,37]. Among them, the Identification Loss usually adopts the Softmax cross-entropy loss function, and treats the people ReID problem as a multi-classification task. That is, assuming that there are N different people, each of which corresponds to a class, the Softmax function is used to convert the predicted classification scores of the people into a probability distribution:(12)$$p_{i}=\frac{e^{Z_{i}}}{\sum_{j}e^{Z_{i}}}$$

The cross-entropy loss function is also utilized to calculate the error between the predicted probability distribution and the actual labels for each individual. After the loss calculation, the model parameters are optimized through gradient descent to ensure that the feature spaces of similar samples of people cluster together [23]. Here, N represents the total number of people categories; *y* in $y_{i}$ represents the real label of people identity, the i−th element of $y_{i}$ (one-hot encoding); and ${\hat{y}}_{l}$ is the i−th element in the probability distribution of the prediction:(13)$$L_{ID}=-\sum y_{i}\log\left({\hat{y}}_{i}\right)$$

Since ID Loss only considers the relationship between samples and class centers, it tends to overlook the relative distances between samples, potentially leading to larger intra-class distances. Therefore, TriLoss is typically used in conjunction with ID Loss to help optimize the relative distances between samples, thereby enhancing the model’s ability to distinguish features. Additionally, the core concept of TriLoss is as follows: for any Anchor sample $x_{a}$ in the graph library, the model aims to minimize the distance between this Anchor and its corresponding positive sample $x_{p}$ (where f(x) represents the vector of the corresponding feature):(14)$$D_{a,p}={\left\|f(x_{a})-f(x_{p})\right\|}_{2}$$

In addition, it is smaller than the distance between the Anchor and the corresponding $x_{N}$ negative sample.(15)$$D_{a,N}={\left\|f(x_{a})-f(x_{N})\right\|}_{2}$$
where at least a threshold (M) between the two should be maintained. This ensures that tight clustering within the class can be formed in the feature space, and the distribution between classes can be controlled to be scattered. In simple terms, the TriLoss is designed to control the relationship between $D_{a,p}+M\leq D_{a,N}$, If this condition is met, the loss is zero; otherwise, the model adjusts its parameters through backpropagation to bring $D_{a,p}$ closer together while pushing $D_{a,N}$ further apart. The formula for calculating $L_{Tri}$ is as follows:(16)$$L_{Tri}=\max(D_{a,p}-D_{a,N}+M,0)$$

ID Loss and TriLoss can be used to enhance the compactness and separability between classes of features, thereby improving the model’s performance of people ReID in complex scenes. Therefore, this paper sets the loss function with weight parameters as follows:(17)$$L=\alpha [L_{ID}+L_{Tri}]f_{patch}+\beta [L_{ID}+L_{Tri}]f_{Token}$$
where L represents the total loss, α represents the loss weight of patch features, and β represents the loss weight of token features. By selecting different weights, the attention to different features can be adjusted.

In Equation (17), the weight parameters are set as *α* = 0.3 and *β* = 0.7 (where *α* denotes the weight for the loss of patch features, and *β* represents the weight for the loss of token features), with their sum being 1 to ensure the stability of the loss scale. The core significance of this setting lies in strengthening the proportion of TriLoss in the total loss through the weight allocation, where *β* > *α*. This is because patch features may have limited capacity for detailing expression due to parameter compression, whereas the global semantics captured by token features rely more on the modeling of relative relationships between samples. TriLoss, thus, can optimize the discriminability of global features.

## 4. Experiments and Discussion

### 4.1. Implementation Details and Datasets

In the verification stage of experimental results, the purpose is to verify the ReID effectiveness of TE-TransReID model on various public datasets and the lightweight parameter number of the model itself. The experiments are carried out on the computer of RTX-4090 (NVIDIA Corporation, Santa Clara, CA, USA) based on Pytorch. The pre-trained model on the ImageNet dataset is used as the backbone network for the model, and all images of people are resized to 256 × 128 as the input. Each training batch contains 64 images, and four of them are sampled for constructing TriLoss. The model after each training is saved and evaluated on the test set. In addition, under the optimizer SDG under stochastic gradient, the base learning rate is set to 1 × 10^−4^, the bias learning rate is twice as high, and L2 regularization is used to avoid overfitting. In this experiment, data comparison is carried out on multiple public people datasets: Market1501, DukeMTMC, and MSMT17 [14,15,16]. Among them, Market1501 is the most commonly used dataset, which contains 32,668 images and 1501 IDs. DukeMTMC contains 16,522 training images and 2228 query images with 1812 IDs. The MSMT17 is the most challenging and complex dataset, comprising 126,441 images of 4101 IDs captured from 15 different cameras. Since all datasets used in this study are multi-camera views and have been trained on TransReID, we incorporated position and camera viewpoint information into the model’s feature processing. The camera ID embedding has a coefficient weight of 3.0 to address domain bias between different cameras. Based on the learning of people ReID in the past, at the mean Average Precision (mAP), rank-1 and rank-5 [19,43], we compare the accuracy and the parameters of each model to reflect the realization of lightweight function.

Thereinto, rank-1 is defined as the proportion of query samples for which the top-ranked sample in the similarity-sorted gallery is of the correct identity. Formally, for each query, the model ranks all gallery samples. A successful match is counted if the sample ranked first shares the same identity as the query. The rank-1 accuracy is the ratio of these successful matches to the total number of queries, serving as a direct indicator of the model’s performance when only the single most similar result is considered. Similarly, rank-5 measures the proportion of query samples where the correct identity is found within the top 5 positions of the ranked list. It is calculated as the ratio of queries for which at least one of the first five ranked samples is a true match to the total number of queries [44]. The mAP provides a more comprehensive evaluation by considering the performance across all recall levels. Its calculation involves two main steps. First, for each query sample, the Average Precision is computed. AP summarizes the precision–recall curve for a single query and is calculated as the average of precision values at each rank where a relevant (correct) sample is retrieved. Second, the mAP is obtained by taking the mean of the Average Precision scores across all query samples. This metric represents a weighted average of precision values at different levels of recall, offering a holistic view of the model’s retrieval performance.

### 4.2. Ablation and Model Selection

In order to realize the lightweight of the model, we first verify the effectiveness of the feature-fusion block of the design, and then select the Transformer branch model. Among them, the fusion fast experiment is to verify the effectiveness of the designed ETFFM and EPFFM. The branch experiment is mainly to experiment with the type and number of layers of the ViT model, and observe the final accuracy and parameter quantity of each model, so as to find the balance point. Here is a demonstration of the two experiments:

(1) Effect of Feature Fusion: Ablation experiments were conducted on the two components of the feature-fusion module, ETFFM and EPFFM, to verify the effectiveness and lightweight nature of the proposed fusion algorithm. Performance analysis was performed on ETFFM and EPFFM using experimental data from Table 1. As indicated by the data in Table 1, the feature-extraction network exhibits significantly degraded performance on the ReID task when both ETFFM and EPFFM are excluded. This is because the model only extracts features from each region separately at this time, and the lack of fused fragmentation information will lead to the lack of diversity in image processing. Therefore, this paper highlights the effectiveness of ETFFM and EPFFM from the perspective of various accuracy changes. As shown in Table 1, mAP, rank-1 and rank-5 increased by 11.3%, 5.8%, and 6.9%, respectively, after adding ETFFM blocks compared with those without adding fusion modules. After the addition of EPFFM blocks, they increased by 7.1%, 3.5%, and 4.6%, respectively. The ablation experiments also show that only integrating token or patch features will also affect the overall performance of the model.

To validate the generalizability of the proposed method, we extended the ablation experiments to the DukeMTMC dataset. The results show that on both Market1501 and DukeMTMC datasets, when both fusion modules are removed simultaneously, the model performance significantly degrades (with an mAP of 55.6% and a rank-1 accuracy of 75.8%). After adding only ETFFM, the mAP, rank-1, and rank-5 metrics increase by 11.8%, 4.3%, and 5.1%, respectively. Upon further incorporating EPFFM, these three metrics rise again by 6.9%, 7.8%, and 8.7%, ultimately reaching an mAP of 74.3% and a rank-1 accuracy of 87.9%. This trend is consistent with the experimental results on Market1501: ETFFM strengthens token-level feature interaction through a gating mechanism, effectively bridging the semantic gap between global and local features on both datasets; EPFFM, leveraging a lightweight Transformer to aggregate patch-level information, further enhances the model’s ability to capture fine-grained differences. The consistent improvements across datasets indicate that the designed fusion modules are not limited to a single scenario but can stably enhance the comprehensiveness and discriminability of feature fusion, thus verifying the generalizability of TE-TransReID under different data distributions.

The above experiments use the basic two-layer ViT-base model, and the basic experiments are carried out to verify the feasibility of feature-fusion processing.

(2) The ViT model selection experiment was carried out on the Transformer branch: The model selection and model parameters of different branches under the overall TE-TransReID network framework will still have a certain impact on the network performance. In addition, the framework is based on the CNN–Transformer double-branch network, and under the reference TransReID method, the Jigsaw Patch Module (JPM) is considered for the global and local information feature blocks of the two branches [11], which increases the robustness of the model by disrupting the order of the original feature information. The additional perturbations introduced do not degrade the performance of the model, but rather improve its ability to capture and reassemble features in training, i.e., the model proposed in this paper contains JPM by default. According to the data in Table 2, the accuracy of the model will be different when different ViT models are selected, and the accuracy of mAP and rank-1 will improve with the accumulation of the number of layers of the model.

Therefore, through comparison and ablation experiments, we chose “ViT-Small 4 *” as the Transformer branching model after trade-off the accuracy and model parameters. The parameters of the double-branch model also achieve the purpose of lightweight under certain conditions, which further verifies the feasibility and superiority of the proposed model.

### 4.3. Comparison with the State of the Art

In Table 3, a comparison with state-of-the-art frameworks on the above three datasets is shown, mainly focusing on map and rank-1 data as a reference for accuracy comparison. We compare various types of people ReID frameworks in recent years, including CNN-based, Transformer, and hybrid networks. The comparison of different basic frameworks reflects the difference in the feasibility of the proposed model algorithm compared to the accuracy. Then, the comparison of parameters further reflects the superiority of the TE-TransReID method in this paper for the dual guarantee of lightweight performance and matching accuracy.

Finally, in order to show the implementation of the model function in this paper more intuitively, we visualized and analyzed the public dataset used in the experiment. First of all, the visualization result is determined by the retrieval of ten ReID samples of peoples to be queried, and from the comparative results in Table 3, it can be seen that different types of person re-identification methods exhibit distinct strengths and limitations. CNN-based methods (e.g., MGN, BFE) achieve high accuracy on medium-scale datasets—for instance, MGN attains a rank-1 accuracy of 95.7% on Market1501—with relatively manageable parameter sizes (30.9 M–42.8 M). However, constrained by the local receptive field of convolutional operations, they perform poorly in occluded scenarios and struggle with modeling global semantic associations, showing insufficient adaptability to complex datasets like MSMT17 [49,50]. Transformer-based methods (e.g., TransReID, AAformer) maintain advantages in cross-camera matching and occlusion handling. Nevertheless, their large parameter sizes (86.1 M–92.4 M) result in high computational costs, making them difficult to deploy in resource-constrained scenarios. Hybrid models (e.g., FusionReID, ABDNet) integrate CNNs’ local details with Transformers’ global semantics, achieving breakthroughs in accuracy—FusionReID, for example, reaches 95.9% rank-1 on Market1501. However, their complex dual-branch designs lead to a sharp surge in parameters (129.5 M–153.8 M), significantly increasing training difficulty and inference latency. In contrast, our proposed TE-TransReID maintains accuracy comparable to the aforementioned methods while compressing parameters to 27.5 M through a lightweight dual-branch structure and targeted fusion modules. It avoids the shortcomings of pure CNNs in global modeling and addresses the issues of parameter redundancy and heavy computation in traditional Transformers and hybrid models, achieving a superior balance between accuracy and efficiency.

We use red to indicate the error retrieval, which can be seen from the similar dresses in Figure 6 and Figure 7, and different rays may cause the recognition and matching of some wrong results.

Finally, as can be seen from the display of the ReID results in the figure, the model in this paper can basically achieve the ReID task under multiple datasets. Although similar people may cause very few recognition errors under different lighting and shading conditions, in terms of the overall effect, TE-TransReID can realize the ReID task on the basis of being a lightweight model.

## 5. Conclusions

In this paper, the proposed TE-TransReID framework leverages a lightweight CNN–Transformer dual-branch architecture combined with an efficient feature-fusion module, demonstrating a balanced trade-off between recognition accuracy and computational efficiency. Nevertheless, certain limitations remain. For instance, in highly challenging scenarios—such as severe pedestrian occlusion, adverse lighting conditions, or motion-induced blur—the limited capacity of lightweight networks to capture fine-grained local features leads to a more pronounced performance drop compared to models with larger parameter scales. Furthermore, when applied to ultra-large-scale datasets, the model’s generalization capability and training efficiency still require improvement, and the potential of its lightweight design for fully capturing diverse feature representations has yet to be fully realized. Future work will address these issues by exploring more robust feature-extraction mechanisms and developing lightweight optimization strategies tailored for large-scale data settings.

## Figures and Tables

**Figure 1 sensors-25-05461-f001:**
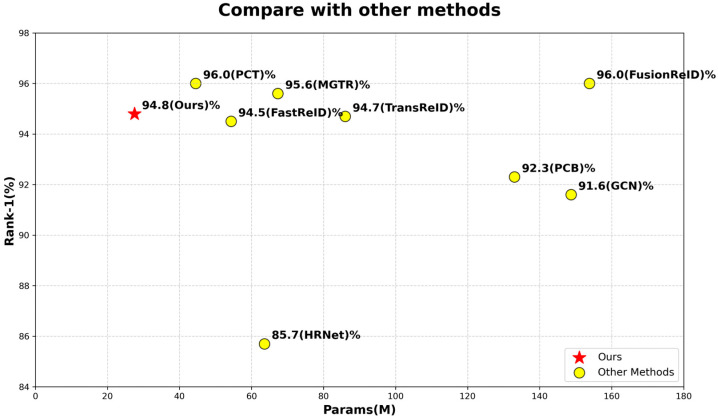
Comparison of TE-TransReID and other models on Market1501 dataset. The “red star” represents the methods used in this article, and the “yellow dots” represent the other methods.

**Figure 2 sensors-25-05461-f002:**
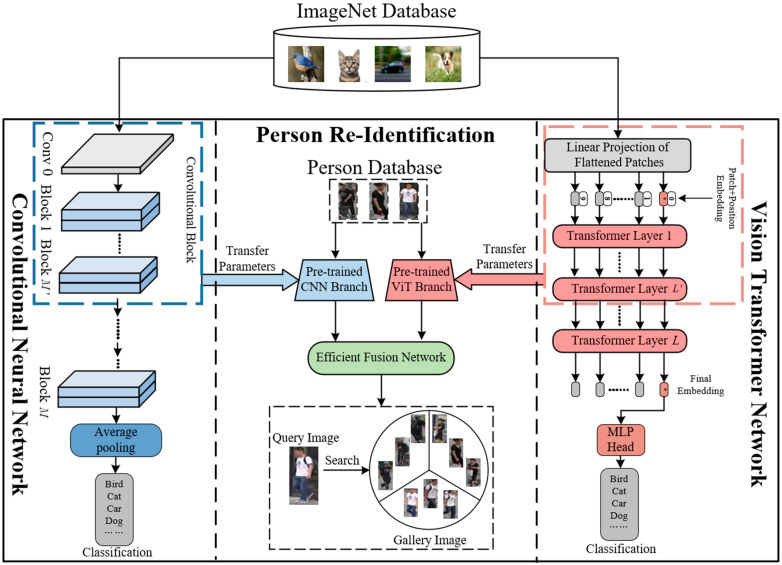
The schematic diagram of the overall framework. The main framework network uses lightweight models pre-trained by ImageNet Database. The (**left**) is CNN framework, which uses MobileNetV2 architecture. On the (**right**) is the Vision Transformer architecture. The two branches jointly process the person dataset, and then fuse the features to realize the ReID function of persons.

**Figure 3 sensors-25-05461-f003:**
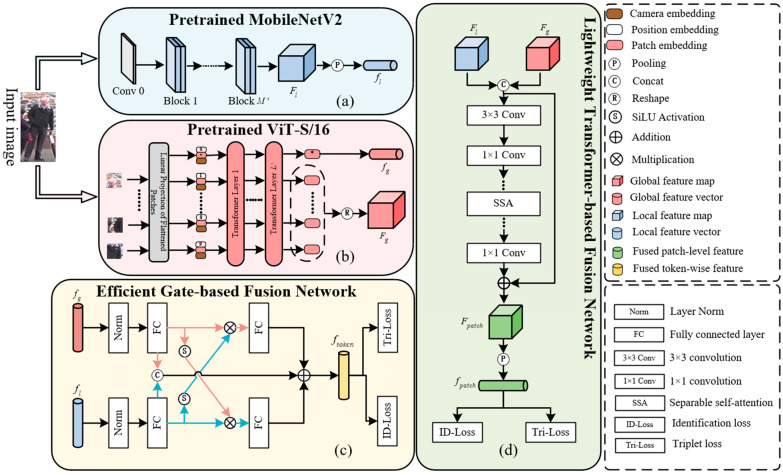
The overall architecture of the proposed TE-TransReID. (**a**) MobileNetV2 is used as the local feature-extraction branch network, (**b**) Vision Transformer is used as the global feature-extraction branch network, (**c**) fuse features in the form of tokens, (**d**) fuse features in the form of patches. The entire architecture jointly realizes the ReID function of human beings.

**Figure 4 sensors-25-05461-f004:**
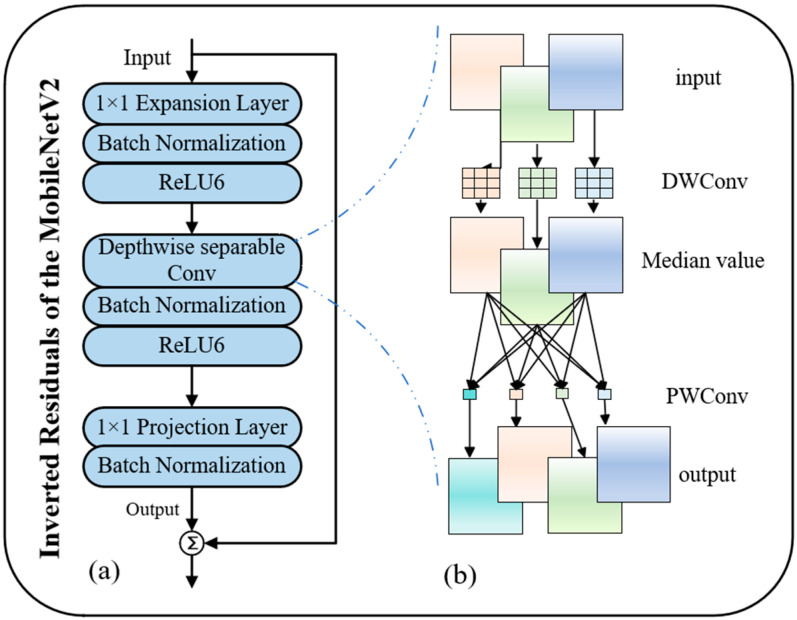
(**a**) The residual structure of the MobileNetV2 model; (**b**) workflow diagram of deeply separable convolutions. Among them, the depth separable convolution blocks are divided into depthwise convolution (DWConv) and pointwise convolution (PWConv).

**Figure 5 sensors-25-05461-f005:**
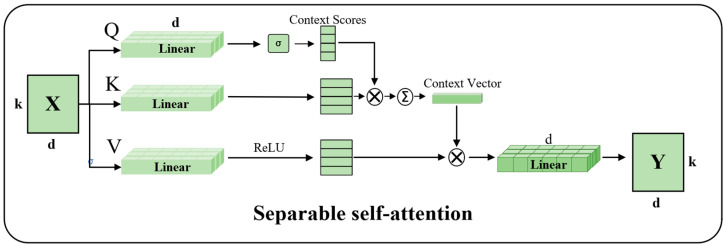
The schematic diagram of the separable self-attention mechanism. The matrix *X* represents the input features with dimension k × d, which first undergoes linear transformations to generate the query matrix *Q*, key matrix *K*, and value matrix *V*. Context scores are computed to characterize the correlation between different positions, while the context vector—serving as the key semantic base vector that the model needs to focus on—is generated simultaneously. Subsequently, features at different positions are weighted and summed according to their relevance. Finally, the context vector activated by ReLU is element-wise multiplied with the value matrix *V* to further obtain the output feature Y.

**Figure 6 sensors-25-05461-f006:**
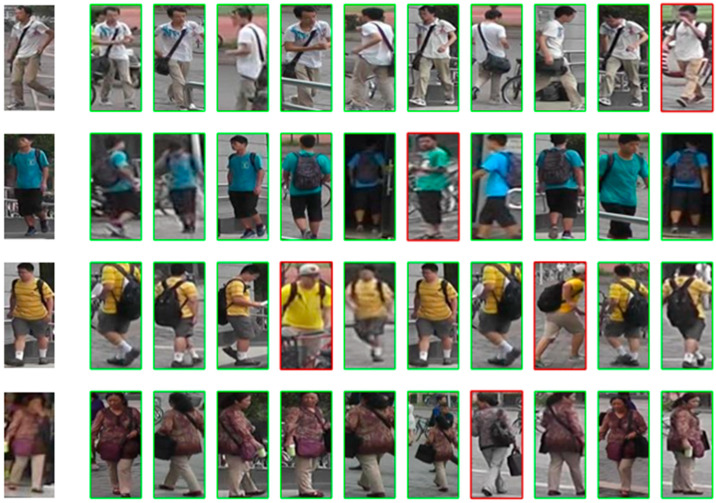
The TE-TransReID search results on the Market1501 dataset. The left side of the image is the image to be queried, and the right side shows the top ten search results. An image in a green box indicates a correct match, while an image in a red box indicates a mismatch.

**Figure 7 sensors-25-05461-f007:**
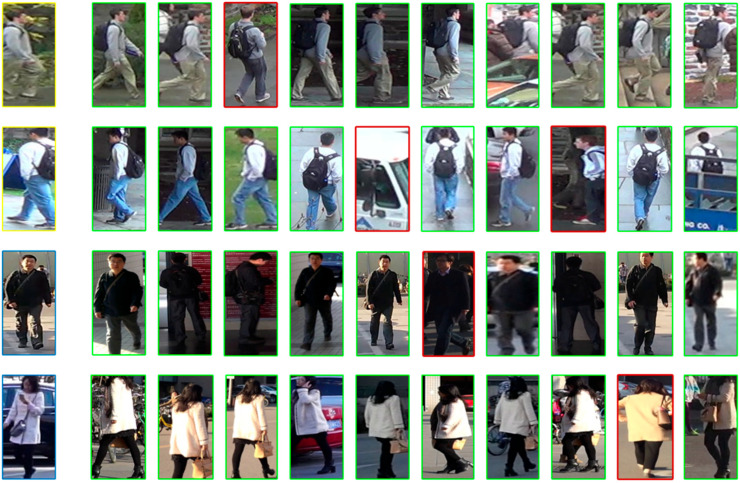
TE-TransReID search results on the DukeMTMC and MSMT17 datasets. The “yellow border” represents a partial visualization of the DukeMTMC dataset. The “blue border” represents a partial visualization of the MSMT17 dataset. The meanings of the red and green borders are the same as those in Figure 6.

**Table 1 sensors-25-05461-t001:** Ablation experiments of the fusion module on Market1501 and DukeMTMC. The two lightweight feature-fusion blocks designed in this paper were ablated separately to reflect the importance of global and local feature-fusion processing from the data.

ETFFM	EPFFM	Market1501			DukeMTMC		
		mAP	Rank-1	Rank-5	mAP	Rank-1	Rank-5
×	×	61.2	81.5	85.5	55.6	75.8	80.3
√	×	72.5	87.2	92.4	67.4	80.1	85.4
√	√	79.6	90.7	97.0	74.3	87.9	94.1

**Table 2 sensors-25-05461-t002:** The implementation of ablation in the model branches. Experiments were conducted using MobilenetV2 and ViT models at different scales to explore models that would strike a balance between lightweight parameters and ReID performance. In the table, “s” represents the small model, “b” represents the base model, and the “*” represents the number of layers of the model.

ViT Model	mAP (%)	Rank-1 (%)	Rank-5 (%)	Params (M)
ViT-s 2 *	79.6	90.7	97.0	16.30
ViT-s 4 *	87.7	94.8	96.8	27.50
ViT-s 8 *	89.9	95.0	97.6	50.00
ViT-b 2 *	79.7	89.4	96.2	18.55
ViT-b 4 *	84.0	91.3	96.9	32.00
ViT-b 6 *	85.7	93.4	97.4	45.50
ViT-b 8 *	85.9	92.7	97.2	59.00

**Table 3 sensors-25-05461-t003:** A comparison of the proposed method with other networks on commonly used public datasets, where “C” stands for CNN-based network, and “T” stands for Transformer-based network. Bold indicates our frame-of-reference model data, and underlined values indicate the optimal result; “r-1” indicates rank-1 precision.

Method	Frame	Params (M)	Market1501		DukeMTMC		MSMT17	
			mAP	r-1	mAP	r-1	mAP	r-1
MGN [18]	C	37.2	86.9	95.7	78.4	88.7	-	-
BFE [45]	C	30.9	86.2	95.3	75.9	88.9	51.5	78.8
HOReID [5]	C	42.8	84.9	94.2	75.6	86.9	-	-
CDNet [46]	C	34.6	86.0	95.1	76.8	88.6	54.7	78.9
FastReID [8]	C	54.3	88.2	94.5	82.3	79.8	59.9	83.3
TransReID [11]	T	86.1	88.9	94.7	82.0	90.7	67.4	85.3
AAformer [47]	T	92.4	87.7	95.4	80.0	90.1	62.6	83.1
PFD [41]	T	90.6	89.7	95.5	-	-	64.4	83.8
ABDNet [48]	C + T	129.5	88.3	95.6	78.6	89.0	60.8	82.3
NFomer [42]	C + T	133.2	91.1	94.7	83.5	89.4	59.8	77.3
FusionReID [24]	C + T	153.8	90.9	9 6.0	82.9	90.3	68.3	86.1
**Ours**	**C + T**	** 27.5 **	**87.7**	**94.8**	**80.6**	**88.3**	**66.2**	**85.7**

## Data Availability

The original contributions presented in the study are included in the article; further inquiries can be directed to the corresponding author.

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
