# Peer review of "TE-TransReID: Towards Efficient Person Re-Identification via Local Feature Embedding and Lightweight Transformer"

_sensors, 2025, doi:10.3390/s25175461_

Round 1
Reviewer 1 Report
Comments and Suggestions for Authors
This paper proposes an efficient person re-identification framework, TE-TransReID. However, the description of methodological innovation is insufficient, experimental comparisons are incomplete, and formatting issues require further refinement. Notably, there are significant formatting errors in the in-text citations of the references; authors are advised to thoroughly review the manuscript before resubmission. To enhance the paper’s completeness and persuasiveness, it is recommend supplementing theoretical derivations, comparative experiments, and limitations analysis.
- The core innovations and comparative advantages over existing methods are not explicitly stated in the abstract.
- The introduction does not clearly define the issues of "lightweight" and "high accuracy" that this paper addresses.
- Supplement the derivation details of the formulas and clarify the definitions of the symbols.
- Ablation experiments need to include results from other datasets and analysis of the impact of different modules to validate the generalizability of the method.
- The conclusion does not clearly point out the limitations of the proposed method.
- The reference format is inconsistent.
Reviewer 2 Report
Comments and Suggestions for Authors
Comments and Suggestions
The reviewed manuscript does only partially meets expectations in accordance with the announced intentions. Comments is described below:
Note 1:
Fig.5 presented set of the three vectors, namely (Q, K, V). However, from the commentary to this figure, appears that such set is presented by (S,K,V) where:
- S (Generate Context Scores) represents the correlation between different positions,
- K model (Context Vector) focus on key semantic base Vector
- V value matrix (Context vector) activated by ReLU
Question: What set of three vectors is true?
Note 2:
Remark. There are repetitions in the manuscript text that should be eliminated. E.g.
Line 31: Efficient Patch-based Feature Fusion Module (EPFFM)
Line 254: Efficient Patch based Feature Fusion Module (EPFFM)
one more:
Line 29: Efficient Token-based Feature Fusion Module (ETFFM)
Line 305: Efficient Token-based Feature Fusion Module (ETFFM)
and one more:
Line 182: self-attention mechanism (SSA)
Line 268: Self-Attention mechanism module (SSA).
Note 3:
The article is prepared without due diligence, as evidenced by numerous comments in the manuscript text, i.e.:
- Lines 45/ 46:……as to realize the comparison of person identity across scenes or under different poses in the same scene Error! Reference source not found.
- Lines 62/ 63: In recent years, researchers have constructed a large number of CNN89 Error! Reference source not found and Transformer 1012Error! Reference source not found.
- Lines 64/ 65: …..and used large-scale public person image datasets: Market150113, DukeMTMC Error! Reference source not found.
- Lines 121/ 122/: Spindle Net, PPCL and DSA-reIDand other networks realize the learning of the correspondence between image patches25,Error! Reference source not found., Error! Reference source not found.
- Lines 148/ 149: So this paper designs a parallel processing person ReID network based on MobileNetV2 and Vision Transformer after a lot of learning of the former40Error! Reference source not found..
- Lines 227/ 228: Transformer-based network framework. Learning from the information supplement of methods such as 233135, this paper also adopts a similar information processing.
- Lines 242/ 243/: In addition, for lightweight architecture design, TE-TransReID does not process all image data in all layers of the selected model as usual methods Error! Reference source not found
- Lines 256/ 257 / 258: in this paper, the Patch level features using feature fusion network architecture based on the Transformer1823Error! Reference source not found.
- Lines 278 / 279: operation and K represents the generation of context vectors Error! Reference source not found.
- Lines 332/ 333: Identification Loss and Triplet Loss are used3Error! Reference source not found.
Note 4.
There is a certain discontinuity in the names and terms used in the text
Example 1:
Line 182: self-attention mechanism (SSA). However, below is:
Lines 268/ 269: separable Self-Attention mechanism module (SSA).
Example 2:
Line 261: (Fig. 5) S - represent the correlation between different positions. However, below is:
Line 278: S - represents the correlation between the different contextual semantics
Example 3:
Lines 262/263- K model need to focus on key semantic base Vector, and the characteristics of different position according to its relevance to the weighted sum. However, below is:
Line 278: K represents the generation of context vectors
General remark : Unfortunately, such examples apply to the entire text
Resume
- The manuscript shows promise, but some corrects need before it can be published.
- Conclusions are too general and don’t reflect the research achieved. The conclusion should be an objective summary of the most important findings in response to the specific research questions. A good conclusion might states the key arguments and counterpoint, and d coherent synthesis of the results.
- The manuscript cites don’t a full article list published in journal Sensors, i.e.:
- A Generative Approach to Person Reidentification by Andrea Asperti, Salvatore Fiorilla and Lorenzo Orsini. Sensors 2024, 24(4), 1240; https://doi.org/10.3390/s24041240 - 15 Feb 2024
- Approaches to Improve the Quality of Person Re-Identification for Practical Use by Timur Mamedov, Denis Kuplyakov and Anton Konushin. Sensors 2023, 23(17), 7382; https://doi.org/10.3390/s23177382 - 24 Aug 2023
- Discriminatively Unsupervised Learning Person Re-Identification via Considering Complicated Images by Rong Quan, Biaoyi Xu and Dong Liang Sensors 2023, 23(6), 3259; https://doi.org/10.3390/s23063259 - 20 Mar 2023
- Person Re-Identification with Improved Performance by Incorporating Focal Tversky Loss in AGW Baseline by Shao-Kang Huang, Chen-Chien Hsu and Wei-Yen Wang. Sensors 2022, 22(24), 9852; https://doi.org/10.3390/s22249852 - 15 Dec 2022
- Unsupervised Person Re-Identification with Attention-Guided Fine-Grained Features and Symmetric Contrast Learning by Yongzhi Wu, Wenzhong Yang and Mengting Wang. Sensors 2022, 22(18), 6978; https://doi.org/10.3390/s22186978 - 15 Sep 2022
Recommendation
The revised manuscript version ID sensors-3800837: “TE-TransReID: Towards Efficient Person Re-identification via Local Feature Embedding and Lightweight Transformer” may appear in the journal “Sensors” after carefully done corrects and replenishment.
Author Response
|
Comments 1: [Fig.5 presented set of the three vectors, namely (Q, K, V). However, from the commentary to this figure, appears that such set is presented by (S, K, V) where: S (Generate Context Scores) represents the correlation between different positions, K model (Context Vector) focus on key semantic base Vector, value matrix (Context vector) activated by ReLU. Question: What set of three vectors is true?] |
|
Response 1: We appreciate the reviewer's valuable comments and affirmation of our work, and we have carefully responded the manuscript based on your suggestions. [The (Q, K, V) represents the true vector in the SSA module, where the "S" in the original text is the Context Scores to characterize the correlation between different positions, and is used to generate the Context vector.] We have also made further additions to the variable relationships that were not clearly explained in the first edition of the manuscript. The revised content is as follows: [Schematic diagram of the separable self-attention mechanism. The matrix X represents the input features with dimension k×d, which first undergoes linear transformations to generate the query matrix Q, key matrix K, and value matrix V. Context Scores are computed to characterize the correlation between different positions, while the Context Vector—serving as the key semantic base vector that the model needs to focus on—is generated simultaneously. Subsequently, features at different positions are weighted and summed according to their relevance. Finally, the context vector activated by ReLU is element-wise multiplied with the value matrix V to further obtain the output feature Y.] Where in the revised manuscript this change can be found: [page 9 Line336 to page 10 Line344].
|
|
Comments 2: [There are repetitions in the manuscript text that should be eliminated. Line 31: Efficient Patch-based Feature Fusion Module (EPFFM) Line 254: Efficient Patch based Feature Fusion Module (EPFFM) one more: Line 29: Efficient Token-based Feature Fusion Module (ETFFM) Line 305: Efficient Token-based Feature Fusion Module (ETFFM) and one more: Line 182: self-attention mechanism (SSA) Line 268: Self-Attention mechanism module (SSA).] |
|
Response 2: Agree. Therefore, we have [finished cutting redundant expressions to streamline our article]. Only the first full name is retained in the expression repeated throughout the text, and the other parts are replaced by abbreviations. Where in the revised manuscript this change can be found: [page 9, and Line332], [page 10, and Line358], [page 11, and Line393]
Comments 3: [The article is prepared without due diligence, as evidenced by numerous comments in the manuscript text, i.e.: 1. Lines 45/ 46: ……as to realize the comparison of person identity across scenes or under different poses in the same scene Error! Reference source not found. 2. Lines 62/ 63: In recent years, researchers have constructed a large number of CNN89 Error! Reference source not found and Transformer 1012Error! Reference source not found. Lines 64/ 65: ……and used large-scale public person image datasets: Market150113, DukeMTMC Error! Reference source not found. 1. Lines 121/ 122/: Spindle Net, PPCL and DSA-reID and other networks realize the learning of the correspondence between image patches25, Error! Reference source not found., Error! Reference source not found. 2. Lines 148/ 149: So this paper designs a parallel processing person ReID network based on MobileNetV2 and Vision Transformer after a lot of learning of the former40Error! Reference source not found. 3. Lines 227/ 228: Transformer-based network framework. Learning from the information supplement of methods such as 233135, this paper also adopts a similar information processing. Lines 242/ 243/: In addition, for lightweight architecture design, TE-TransReID does not process all image data in all layers of the selected model as usual methods Error! Reference source not found Lines 256/ 257 / 258: in this paper, the Patch level features using feature fusion network architecture based on the Transformer1823Error! Reference source not found. 1. Lines 278 / 279: operation and K represents the generation of context vectors Error! Reference source not found. 2. Lines 332/ 333: Identification Loss and Triplet Loss are used3Error! Reference source not found.] Response 3: We thank the reviewer for the pertinent comments and affirmation of our work, and we have meticulously revised the manuscript in line with your suggestions. Causing a large number of "Error! Reference source not found" is the domain code that is not locked and cross-referenced when submitting the first version of the manuscript file docx. Now we have [modified the references one by one and set the domain code to ensure that the references are properly referenced and displayed]. The revised content is as follows: [……thereby enabling identity comparison of individuals across scenarios or under varying poses within the same scene[1,2]] [In recent years, researchers have constructed a large number of CNN[8,9,10] and Transformer[11,12,13]……] [……and used large-scale public person image datasets :Market1501[14], DukeMTMC[15], MSMT17[16]] [ ……Spindle Net, PPCL and DSA-reID and other networks realize the learning of the correspondence between image patches[25,26,27,28].] [So this paper designs a parallel processing person ReID network based on MobileNetV2 and Vision Transformer after a lot of learning of the former[40,41].] [Transformer-based network framework. Learning from the information supplement of methods such as[21,31,35]……] [……TE-TransReID does not process all image data in all layers of the selected model as usual methods[10,16,32]] [……in this paper, the Patch level features using feature fusion network architecture based on the Transformer[18,23,36]] [……indicates a zoom operation and K represents the generation of context vectors[41]] [……the traditional Identification Loss and TriLoss are used[3,27,40]] Where in the revised manuscript this change can be found: [page 2, and Line46-47], [page 2, and Line76], [page 2, and Line77], [page 2, and Line78], [page 4, and Line134-135], [page 5, and Line168-169], [page 8, and Line297-298], [page 9, and Line334-335], [page 10, and Line366-367], [page 12, and Line420-421].
Comments 4: [There is a certain discontinuity in the names and terms used in the text. Example 1: Line 182: self-attention mechanism (SSA). However, below is: Lines 268/ 269: separable Self-Attention mechanism module (SSA). Example 2: Line 261: (Fig. 5) S-represent the correlation between different positions. However, below is: Line278: S - represents the correlation between the different contextual semantics Example 3: Lines 262/263- K model need to focus on key semantic base Vector, and the characteristics of different position according to its relevance to the weighted sum. However, below is: Line 278: K represents the generation of context vectors] Response 4: We thank the reviewer for the helpful comments and recognition of our work, and we have carefully revised the manuscript in accordance with your suggestions. We [rephrase the unclear or conflicting parts of the text] to enhance the coherence and fluency of the article's logic and sentences. First, we only [assign the name "SSA" to the Self-Attention mechanism module], and it is inaccurate to express the first self-attention mechanism as SSA. Secondly, in Fig.5, our original ["S" stands for Context Scores, which is used to generate Context Vectors], and we have also modified the problem of unclear expression. Finally, the ["K" is the key matrix in our SSA module, which is used to generate a Context Vector in collaboration with Context Scores]. These are all the answers we have to ask about the reviewers. The revised content is as follows: [After the convolution operation, the features are directly entered into the Separable Self-Attention mechanism module (SSA).] [Figure 5. Schematic diagram of the separable self-attention mechanism. The matrix X represents the input features with dimension k×d, which first undergoes linear transformations to generate the query matrix Q, key matrix K, and value matrix V. Context Scores are computed to characterize the correlation between different positions, while the Context Vector—serving as the key semantic base vector that the model needs to focus on—is generated simultaneously. Subsequently, features at different positions are weighted and summed according to their relevance. Finally, the context vector activated by ReLU is element-wise multiplied with the value matrix V to further obtain the output feature Y.] Where in the revised manuscript this change can be found: [page 10, and Line356-357], [page 9-Line337 to page 10-Line344].
|

Reviewer 3 Report
Comments and Suggestions for Authors
1.Please provide a more detailed explanation of the significance of each architecture in every figure or chart to enhance clarity.
2.There are multiple instances of “Error! Reference source not found.” in the manuscript, making it impossible to verify whether the citations are correct.
3.Please explain the differences between Local Feature Extraction and Global Feature Extraction.
4.In Section 4, please add a discussion to explore the differences among the models presented in this paper, as well as their respective advantages and disadvantages.
Author Response
|
Comments 1: [Please provide a more detailed explanation of the significance of each architecture in every figure or chart to enhance clarity.] |
||||||||||||||||||||||||||||||||||||||
|
Response 1: We are grateful for the reviewer's constructive comments and recognition of our work, and we have earnestly revised the manuscript as per your suggestions. We have [updated the module labels in Fig. 3, and described point-by-point how each branch of the overall architecture and the feature fusion module process features]; additionally, we have summarized the functions and structure of the overall framework. [In Fig. 4, we have provided more explicit descriptions of the changes in feature channels and features regarding the architecture of the depth-wise separable network]. Furthermore, we have [uniformly revised the previously controversial descriptions in Fig. 5 and analyzed the handling of each variable]. Finally, to enrich the paper, we have [supplemented ablation experiments on the DukeMTMC dataset and conducted generalization analysis and verification]. All the above supplementary explanations have made the presentation and supporting evidence of the paper more comprehensive and three-dimensional. The revised content is as follows: 1.[ The functionalities of each component in Figure 3 are elaborated as follows: (a) The pretrained MobileNetV2 branch serves as a lightweight convolutional feature extractor, outputting the local feature map FMlocal​ and the local feature vector fMlocal to capture fine-grained visual details. (b) It converts the input image into a sequential representation, leverages multi-layer Transformers to model long-range dependencies, and outputs the global feature map ​FLglobal and the global feature vector fLglobal ​, thereby strengthening the modeling of semantic correlations. (c) The module in (c) targets vector-level feature fusion: ​fMlocal and fLglobal​ are first normalized, then fed into fully connected layers coupled with a gating mechanism that dynamically learns fusion weights between global and local features. The resultant fused feature is optimized via IDLoss and TriLoss. (d) The module in (d) enables feature map-level fusion: it concatenates ​FMlocal and FLlocal​, incorporates camera embeddings to supplement domain-specific information, aggregates patch-level correlations using Separable Self-Attention, and refines features through a convolution-residual structure, with dual losses (IDLoss and TriLoss) guiding optimization. This architecture establishes feature complementarity via the “convolution (local) + Transformer (global)” dual-branch design, achieves multi-granularity feature aggregation through the “gated vector fusion + lightweight Transformer-based feature map fusion” dual-path mechanism, and enhances domain adaptability via camera embeddings. Ultimately, it generates more comprehensive and discriminative feature representations for person re-identification, while ensuring efficiency and generalizability through lightweight components.] 2.[As illustrated in Figure 4, the Inverted Residual structure of MobileNetV2, breaks away from the design paradigm of traditional residuals and adopt a new way of working: Input features (with M channels) first undergo convolution-based dimension expansion (with a dimension expansion ratio t, resulting in tM channels). This dimension augmentation prevents premature compression of features, thereby better preserving the richness of fine-grained information. Subsequently, a depthwise separable convolution is incorporated, which only models spatial correlations within individual channels, significantly reducing computational overhead. Finally, a 1×1 convolution is used to reduce the dimension back to M (ensuring consistent input and output channel counts to satisfy residual connections). By enhancing feature expression through dimension expansion and optimizing efficiency via depthwise separable convolution, this design enables MobileNetV2 to extract high-quality local features at a lightweight cost, laying a foundation for efficient feature extraction in the convolutional branch of TE-TransReID and supporting the performance of subsequent multi-module fusion] 3.[Figure 1. Schematic diagram of the separable self-attention mechanism. The matrix X represents the input features with dimension k×d, which first undergoes linear transformations to generate the query matrix Q, key matrix K, and value matrix V. Context Scores are computed to characterize the correlation between different positions, while the Context Vector—serving as the key semantic base vector that the model needs to focus on—is generated simultaneously. Subsequently, features at different positions are weighted and summed according to their relevance. Finally, the context vector activated by ReLU is element-wise multiplied with the value matrix V to further obtain the output feature Y. Let the input feature matrix be X (where k denotes the length of the feature sequence and d represents the feature dimension). First, linear transformations are applied to generate the query matrix, key matrix, and value matrix (with, , as learnable linear transformation weights, and d as the transformed dimension). Then, context scores S are computed, modeling the correlation strength between different positions via scaled dot-product attention. Subsequently, a context vector is generated by performing a weighted sum of the value matrix based on the context scores, focusing on key semantic information. Finally, after enhancing nonlinear expression through the ReLU activation function, an element-wise multiplication is performed with the value matrix V to obtain the output feature Y.] 4.[To validate the generalizability of the proposed method, we extended the ablation experiments to the DukeMTMC dataset. The results show that on both Market1501 and DukeMTMC datasets, when both fusion modules are removed simultaneously, the model performance significantly degrades (with an mAP of 55.6% and a rank-1 accuracy of 75.8%). After adding only ETFFM, the mAP, rank-1, and rank-5 metrics increase by 11.8%, 4.3%, and 5.1% respectively. Upon further incorporating EPFFM, these three metrics rise again by 6.9%, 7.8%, and 8.7%, ultimately reaching an mAP of 74.3% and a rank-1 accuracy of 87.9%. This trend is consistent with the experimental results on Market1501: ETFFM strengthens token-level feature interaction through a gating mechanism, effectively bridging the semantic gap between global and local features on both datasets; EPFFM, leveraging a lightweight Transformer to aggregate patch-level information, further enhances the model's ability to capture fine-grained differences. The consistent improvements across datasets indicate that the designed fusion modules are not limited to a single scenario but can stably enhance the comprehensiveness and discriminability of feature fusion, thus verifying the generalizability of TE-TransReID under different data distributions. Table 1. Ablation experiments of the fusion module on Market1501 and DukeMTMC. The two lightweight fusion feature blocks designed in this paper were ablated separately to reflect the importance of global and local feature fusion processing from the data.
] Where in the revised manuscript this change can be found: [page 2,and Line215-231 to page 7, and Line233-239], [page 7, and Line249-262], [page 10, and Line345-354], [page 14, and Line532-546].
|
||||||||||||||||||||||||||||||||||||||
|
Comments 2: [There are multiple instances of “Error! Reference source not found.” in the manuscript, making it impossible to verify whether the citations are correct.] |
||||||||||||||||||||||||||||||||||||||
|
Response 2: Thanks to the reviewers for helping us point out formatting issues with the references, we have [reviewed the citations throughout the article and fixed the error] when converting to pdf due to unlocked domains. Where in the revised manuscript this change can be found: [page 1, and Line47-49, Line68, Line73, Line76-78, Line83-86], [page 2, and Line98, Line122], [page 4, and Line135, Line144-145], [page 5, and Line161-164, Line169, Line178, Line189], [page 7, and Line242], [page 8, and Line298], [page 9, and Line321, Line335], [page 10, and Line367], [page 11, and Line396, Line421], [page 12, and Line430], [page 13, and Line482, Line490], [page 15, and Line557].
Comments 3: [Please explain the differences between Local Feature Extraction and Global Feature Extraction.] Response 3: We appreciate the reviewer's helpful insights and recognition of our work, and we have carefully revised the manuscript according to your suggestions. We have [added a supplement on the differences in global and local feature extraction following Section 2.1&2.2. By comparing the extraction targets and processing methods], this further substantiates the feasibility of our dual-branch network. The revised content is as follows: [In summary, global feature extraction takes the entire image as the processing unit, capturing global semantic correlations and long-range dependencies through encoding the overall information of the image. Local feature extraction, by contrast, focuses on local regions within the image (such as the head, torso, and limbs), capturing fine-grained information through refined processing of local details. The two differ in their extraction scope and the granularity of information they process, which leads to their distinct applicable scenarios. Consequently, in the field of person re-identification, numerous studies currently seek to fuse these two types of features to achieve superior performance.] Where in the revised manuscript this change can be found: [page 4, and Line148 to page 5, and Line155].
Comments 4: [In Section 4, please add a discussion to explore the differences among the models presented in this paper, as well as their respective advantages and disadvantages.] Response 4: We appreciate the reviewer's helpful comments and recognition of our work, and we have carefully revised the manuscript as per your suggestions. In the final experiments, we compared our method with multiple other approaches. Based on the collated experimental data, we [analyzed each model by comparing them in terms of model architecture and parameter classification, examining their respective strengths and weaknesses.] The lightweight and high-precision characteristics of the TE-TransReID method are demonstrated through its performance in accuracy and model parameters. The revised content is as follows: [From the comparative results in Table 3, different types of person re-identification methods exhibit distinct strengths and limitations. CNN-based methods (e.g., MGN, BFE) achieve high accuracy on medium-scale datasets—for instance, MGN attains a rank-1 accuracy of 95.7% on Market1501—with relatively manageable parameter sizes (30.9M–42.8M). However, constrained by the local receptive field of convolutional operations, they perform poorly in occluded scenarios and struggle with modeling global semantic associations, showing insufficient adaptability to complex datasets like MSMT17. Transformer-based methods (e.g., TransReID, AAformer) hold advantages in cross-camera matching and occlusion handling. Nevertheless, their large parameter sizes (86.1M–92.4M) result in high computational costs, making them difficult to deploy in resource-constrained scenarios. Hybrid models (e.g., FusionReID, ABDNet) integrate CNNs’ local details with Transformers’ global semantics, achieving breakthroughs in accuracy—FusionReID, for example, reaches 95.9% rank-1 on Market1501. However, their complex dual-branch designs lead to a sharp surge in parameters (129.5M–153.8M), significantly increasing training difficulty and inference latency. In contrast, our proposed TE-TransReID maintains accuracy comparable to the aforementioned methods while compressing parameters to 27.5M through a lightweight dual-branch structure and targeted fusion modules. It avoids the shortcomings of pure CNNs in global modeling and addresses the issues of parameter redundancy and heavy computation in traditional Transformers and hybrid models, achieving a superior balance between accuracy and efficiency.] Where in the revised manuscript this change can be found: [page 16, and Line589-608].
|
||||||||||||||||||||||||||||||||||||||

Reviewer 4 Report
Comments and Suggestions for Authors
The paper presents a lightweight person reidentification framework that combines pretrained vision transformer and CNN models. The article is well structured and includes literature review, methodology explanation, numerical experiments and analysis. The conclusions meet the raised research questions.
Please find the following comments to improve the quality of the paper.
- please check the referencing technique: there are a lot of "Error! Reference source not found" The existing references also do not meet the citation requirements.
- (176 line) although W, H, C have intuitive meanings of width, height, channels, this should be stated explicitly to avoid misunderstanding.
- (369 line) what is the possible range of \alpha, \beta?
- (Table 1) Consider adding how mAP, rank-1, rank-5 are calculated.
Author Response
|
Comments 1: [Please check the referencing technique: there are a lot of "Error! Reference source not found" The existing references also do not meet the citation requirements.] |
|
Response 1: Thanks to the reviewers for helping us point out formatting issues with the references, we have [reviewed the citations throughout the article and fixed the error] when converting to pdf due to unlocked domains. The revised content is as follows: Display the results of some reference modifications. [……thereby enabling identity comparison of individuals across scenarios or under varying poses within the same scene [1,2]] [In recent years, researchers have constructed a large number of CNN [8,9,10] and Transformer [11,12,13] ……] [……and used large-scale public person image datasets: Market1501[14], DukeMTMC[15], MSMT17[16]] [……Spindle Net, PPCL and DSA-reID and other networks realize the learning of the correspondence between image patches [25,26,27,28].] [So this paper designs a parallel processing person ReID network based on MobileNetV2 and Vision Transformer after a lot of learning of the former [40,41].] [Transformer-based network framework. Learning from the information supplement of methods such as [21,31,35] ……] [……TE-TransReID does not process all image data in all layers of the selected model as usual methods [10,16,32]] [……in this paper, the Patch level features using feature fusion network architecture based on the Transformer [18,23,36]] [……indicates a zoom operation and K represents the generation of context vectors [41]] [……the traditional Identification Loss and TriLoss are used [3,27,40]]
Where in the revised manuscript this change can be found: [page 1, and Line47-49, Line68, Line73, Line76-78, Line83-86], [page 2, and Line98, Line122], [page 4, and Line135, Line144-145], [page 5, and Line161-164, Line169, Line178, Line189], [page 7, and Line242], [page 8, and Line298], [page 9, and Line321, Line335], [page 10, and Line367], [page 11, and Line396, Line421], [page 12, and Line430], [page 13, and Line482, Line490], [page 15, and Line557].
|
|
Comments 2: [(176 line) although W, H, C have intuitive meanings of width, height, channels, this should be stated explicitly to avoid misunderstanding.] |
|
Response 2: We thank the reviewer for the pertinent comments and affirmation of our work, and we have meticulously revised the manuscript in line with your suggestions. To enhance the clarity of variables in the article, we have [supplemented the dimensional meanings of W, H, and C regarding features (where W and H represent the spatial dimensions of the feature map, and C denotes the channel dimension of the feature map)]. This makes the subsequent feature transformations and variable changes in formulas more comprehensible. The revised content is as follows: [(W (Width) and H (Height) indicate the spatial dimensions of the feature map. During the processing of input images by MobileNetV2, the original spatial size of the input image is gradually compressed through convolution, pooling, and other operations. The resulting W and H of the feature map reflect the spatial distribution information of local regions in the image. C (Channel) represents the dimensionality of the feature map in the channel dimension, reflecting the richness of features extracted by the model. In MobileNetV2, the "Inverted Residual Block" is used to expand and compress feature dimensions. The value of C corresponds to the number of channels of the feature after expansion or compression)] Where in the revised manuscript this change can be found: [page 1, and Line28-37].
Comments 3: [(369 line) what is the possible range of \alpha, \beta?] Response 3: We are grateful for the reviewer's constructive comments and recognition of our work, and we have carefully revised the manuscript in accordance with your suggestions. α and β are weight values we designed in the loss function, [with α = 0.3 and β = 0.7 (where α denotes the weight for the loss of patch features, and β represents the weight for the loss of Token features)]. Their sum is 1, which ensures the stability of the loss scale. The revised content is as follows: [In Equation (17), the weight parameters are set as α=0.3 and β=0.7 (where α denotes the weight for the loss of patch features, and β represents the weight for the loss of Token features), with their sum being 1 to ensure the stability of the loss scale. The core significance of this setting lies in strengthening the proportion of TriLoss in the total loss through the weight allocation where β>α. This is because patch features may have limited capacity for detailing expression due to parameter compression, whereas the global semantics captured by Token features rely more on the modeling of relative relationships between samples. TriLoss, thus, can optimize the discriminability of global features.] Where in the revised manuscript this change can be found: [page 13, and Line462-469].
Comments 4: [(Table 1) Consider adding how mAP, rank-1, rank-5 are calculated.] Response 4: We thank the reviewer for the helpful comments and affirmation of our work, and we have carefully revised the manuscript based on your suggestions. We have [supplemented explanations of the evaluation metrics mAP, rank-1, and rank-5] used in the experiments from the perspective of their definitions. Specifically, [mAP refers to the calculation of average precision, which evaluates model performance through the weighted average of queries across different recall levels. Rank-1 and rank-5, on the other hand, refer to the proportion of correctly queried samples in similarity ranking]. These supplements to the evaluation metrics will make our experimental analysis clearer and our arguments more robust. The revised content is as follows: [Thereinto, rank-1 is defined as the proportion of query samples for which the top-ranked sample in the similarity-sorted gallery is of the correct identity. Formally, for each query, the model ranks all gallery samples. A successful match is counted if the sample ranked first shares the same identity as the query. The rank-1 accuracy is the ratio of these successful matches to the total number of queries, serving as a direct indicator of the model’s performance when only the single most similar result is considered. Similarly, rank-5 measures the proportion of query samples where the correct identity is found within the top 5 positions of the ranked list. It is calculated as the ratio of queries for which at least one of the first five ranked samples is a true match to the total number of queries. The mAP provides a more comprehensive evaluation by considering the performance across all recall levels. Its calculation involves two main steps. First, for each query sample, the Average Precision is computed. AP summarizes the precision-recall curve for a single query and is calculated as the average of precision values at each rank where a relevant (correct) sample is retrieved. Second, the mAP is obtained by taking the mean of the Average Precision scores across all query samples. This metric represents a weighted average of precision values at different levels of recall, offering a holistic view of the model’s retrieval performance.] Where in the revised manuscript this change can be found: [page 13, and Line493-509].
|

Round 2
Reviewer 1 Report
Comments and Suggestions for Authors
The authors have addressed all my concerns.
Seems authors' contribution statement, acknowledgement and other statements are missing.
Author Response
Comments: [Seems authors' contribution statement, acknowledgement and other statements are
missing.]
Response: We thank the reviewer for the helpful comments and recognition of our work, and
we have carefully revised the manuscript in accordance with your suggestions. We have added
[supplements to (including but not limited to) acknowledgments, funds, author contributions, etc. at
the end of the article], making our article materials more complete and fully explained.
The revised content is as follows:
[Author Contributions: Xiaoyu Zhang was responsible for Data curation, Methodology,
Software, Validation, Visualization, and Writing – original draft in the research; Ning Jiang
was responsible for Formal analysis and Investigation; Ke Xu was responsible for Formal
analysis and Writing – review & editing; Wenbo Zhu was responsible for Formal analysis and
Supervision; Rui Cai was responsible for Investigation; Yaocong Hu was responsible for
Methodology, Project administration, Validation, and Writing – review & editing; Huicheng
Yang was responsible for Resources and Supervision; Minwen Xing was responsible for Software.
Funding: The National Natural Science Foundation of China (No.62203012), The Natural
Science Foundation of the Anhui Higher Education Institutions of China (No. 2023AH030020)
Institutional Review Board Statement: Not applicable.
Informed Consent Statement: Not applicable.
Data Availability Statement: The original contributions presented in the study are included
in the article, further inquiries can be directed to the corresponding author.
Acknowledgments: The authors would like to thank Shuting He, whose implementation of
the high-precision model in ReID provided great inspiration for this research—their code and
model represented new breakthroughs in the field at that time, enabling the Transformer
model to be widely applied to ReID tasks, and the corresponding author for the guidance,
review and supervision throughout the manuscript, which made this paper more
comprehensive and coherent.
Conflicts of Interest: Authors Xiaoyu Zhang, Ke Xu, Wenbo Zhu, Rui Cai, Yaocong Hu,
Huicheng Yang, and Minwen Xing were employed by Anhui Polytechnic University. The
author Ning Jiang was employed by Northeast Forestry University. The remaining authors
declare that the research was conducted in the absence of any commercial or financial
relationships that could be construed as a potential conflict of interest.]
Where in the revised manuscript this change can be found: [page 18, and Line640-671].

Reviewer 3 Report
Comments and Suggestions for Authors
This manuscript has fully addressed my concerns and is, in my opinion, suitable for publication.
Author Response
|
Comments: [The English could be improved to more clearly express the research.] |
|
Response: We appreciate the reviewer's valuable comments and affirmation of our work, and we have carefully responded the manuscript based on your suggestions. [We have rewritten the language of each chapter to make our article more relevant to the requirements]. The revised content is as follows: [However, their high computational costs and inadequate capacity to capture fine-grained local features impose significant constraints on re-identification performance.] [Ideally, person ReID technology should be capable of addressing the occlusion of individuals, variations in illumination, postural variations, and other challenges present in large volumes of image and video data.] [However, CNNs exhibit a weak ability to model global contextual relationships (struggling to capture associations between distant pixels) and also tend to overemphasize local information, which ultimately results in feature redundancy.] [In other words, numerous existing methods adopt multi-branch network architectures or complex feature fusion modules, which ultimately elevates the difficulty of model training and optimization.] [This framework leverages the complementary strengths of CNN and Transformer, where the CNN network processes local feature information, while the Vision Transformer network handles global feature information.] [It mitigates issues such as large model size and low training/inference speed, which are incurred by stacking deep residual blocks in ResNet—a commonly used CNN backbone for local feature extraction in person ReID.] Where in the revised manuscript this change can be found: [page1 Line22-23], [page2 Line48-50], [page4 Line136-139], [page5 Line166-168 and Line188-189], [page7 Line244-246], [page14 Line523-525]
|
